# Using deep learning to predict outcomes of legal appeals better than human experts: A study with data from Brazilian federal courts

**Elias Jacob de Menezes-Neto**[1⊙]*, **Marco Bruno Miranda Clementino**[2,3⊙]

**1** Department of Law, Universidade Federal do Rio Grande do Norte, Caicó, RN, Brazil, **2** Department of Private Law, Universidade Federal do Rio Grande do Norte, Natal, RN, Brazil, **3** 6th Federal Court, Tribunal Regional Federal da 5a Região, Natal, RN, Brazil

⊙ These authors contributed equally to this work.
* elias.jacob@ufrn.br

**Data Availability Statement:** Data and pretrained models: https://www.kaggle.com/eliasjacob/brcad5 - DOI: 10.34740/kaggle/dsv/3310186 Github repo

## Abstract

Legal scholars have been trying to predict the outcomes of trials for a long time. In recent years, researchers have been harnessing advancements in machine learning to predict the behavior of natural and social processes. At the same time, the Brazilian judiciary faces a challenging number of new cases every year, which generates the need to improve the throughput of the justice system. Based on those premises, we trained three deep learning architectures, ULMFiT, BERT, and Big Bird, on 612,961 Federal Small Claims Courts appeals within the Brazilian 5th Regional Federal Court to predict their outcomes. We compare the predictive performance of the models to the predictions of 22 highly skilled experts. All models outperform human experts, with the best one achieving a Matthews Correlation Coefficient of 0.3688 compared to 0.1253 from the human experts. Our results demonstrate that natural language processing and machine learning techniques provide a promising approach for predicting legal outcomes. We also release the Brazilian Courts Appeal Dataset for the 5th Regional Federal Court (BrCAD-5), containing data from 765,602 appeals to promote further developments in this area.

## Introduction

Legal judgment prediction (LJP) is one of the most common tasks that legal scholars perform every day. Lawyers and their clients try to predict the outcome of lawsuits, hoping to improve their chances of achieving favorable results. Despite being neutral players, judges may also desire to know if a higher court would overrule their decision in case of an appeal, especially in legal systems in which stare decisis must be observed.

The ability to foresee how a conflict will end is deeply tied to the law itself, as a way to make social interactions more predictable, stable, and reliable. Some authors argue that predicting the behavior of courts can enhance our understanding of decision theory and shed light on the evolution of the judiciary [1]. Predicting outcomes of legal problems has high economic, political, and theoretical value.

with code: https://github.com/eliasjacob/paper_
brcad5.

**Funding:** This study was funded by the National
Council for Scientific and Technological
Development (CNPq) through a scholarship to
Elias Jacob de Menezes-Neto (302668/2020-9).
The Brazilian Coordenação de Aperfeiçoamento de
Pessoal de Nível Superior (CAPES) financed the fee
to publish this article (finance code 001). Funders
had no role in the study design, data collection and
analysis, or the decision to prepare and publish the
manuscript.

**Competing interests:** Elias Jacob de Menezes-Neto
declares no competing interests. Marco Bruno
Miranda Clementino is a federal judge in the 5th
Regional Federal Court jurisdiction. Although his
position is not affected in any way by this paper,
we understand that this affiliation may be seen as a
non-financial competing interest. This does not
alter our adherence to PLOS ONE policies on
sharing data and materials.

Unsurprisingly, computers have long been considered a possible tool to aid legal scholars in achieving this goal [2]. More recently, researchers have become interested in harnessing the power of machine learning (ML) to predict the fate of legal cases. Several legal and technological factors support this trend, but, mainly, this advancement is now possible due to the increased adoption of electronic filing systems. Having almost 100% of their cases processed electronically, Brazilian federal courts generate large amounts of data as a byproduct of their daily activities. All parties must submit their filings electronically. Since 2006, in some instances, no paper at all is submitted. This creates large amounts of data available for analysis. In 2019 only, Brazilian Federal Trial Courts received 4,118,969 new lawsuits, of which 73% (3,003,387) were filed at Federal Small Claims Courts (FSCC). During the same year, Federal Appeals Courts received 1,078,049 new appeals, of which almost half (536.048) came from Federal Small Claims Courts. A three-judge appellate panel (AP, also known as "Turma Recursal") hears each appeal and tries the case, usually ending in one of the following results: "full reverse", "partial reverse" or "affirm" the first instance decision. That result from the AP is what we want to predict in this paper.

Appellate panels affirm lower court decisions in 79% of cases. As a consequence of losing, most appellants have to pay legal expenses for the prevailing party, representing a significant economic impact for those seeking to modify previous rulings. From a broader social perspective, there are no real winners when the affirmation rates are so high and steady. On the one hand, if the losing party is the federal government, taxpayer money is lost paying expenses due to an ill-fated appeal. On the other hand, if the government wins, people who are usually already impoverished must pay the government's legal fees, making their financial situation even worse than when they began. As if that weren't enough, unnecessary appeals hinder the capability of the judiciary to handle cases in a timely manner.

The ability to better predict an outcome can lead to better decisions about when to appeal, causing an overall drop in the proportion of decisions that affirm the first instance court ruling. It could also enhance the workflow of the courts by automatically evaluating an appeal and suggesting the best outcome for that specific case. In such cases, an AI could enhance the throughput of the judiciary.

Considering this, we propose the use of natural language processing and deep learning models to predict the outcomes of appeals tried by these APs, showing that our models are not only better than random guessing but, primarily, that they outperform human experts on this task. Using only the final decision text of trial courts, our models outperform by almost three times the predictions determined by human experts.

We evaluated three deep learning architectures, of which one [3] is exclusively based on a classifier layer built on top of a tweaked Long Short-Term Memory (LSTM) network [4]. For the second architecture [5], we used an approach that combines a Transformer, in our case, a BERT-based model [6], with an LSTM as a way to overcome sequence length limitations for traditional Transformers. Finally, the third model uses a Big Bird architecture [7] capable of handling texts with up to 7,680 tokens.

AI-driven systems could potentially improve stability, predictability, efficiency, and fairness in a legal system with more than 80 million new cases every year, strengthening the use of legal precedents by the Brazilian judiciary. On one side, lawyers can leverage machine learning models to reduce litigation costs and legal expenses, leading to broader access to legal services. On the other side, the judiciary can also benefit from this kind of approach by creating automated management systems for most common cases. This technology could potentially improve the throughput of legal systems by supporting federal judges and their staff.

While our models cannot replace human decision-making, they can assist courts in handling more cases. Machine learning solutions would further expand access to justice, especially

for often marginalized groups that commonly seek to assert their rights in Federal Small Claims Courts.

## Institutional background

The Brazilian judiciary is defined in the Constitution, in articles 92 through 126. It consists of several bodies divided into branches structured based on the distribution of jurisdictional powers (federal, state, labor, electoral, and military).

In federal and state courts, there also exists a parallel structure which is found in the small claims courts, with the specific purpose of processing less complex civil and criminal cases. The concept of "less complexity" referred to in the Constitution is interpreted in terms of the economic value involved in the case. Thus, civil lawsuits of lesser economic value are as a rule processed in the small claims courts, where certain procedural privileges are attributed to citizens.

Even though they are part of the federal and state courts, the small claims courts have a completely different organizational structure and form a subsystem within each of the above-mentioned branches. Thus, when an appeal is filed against the decisions of these courts, the judgment is carried out by an appellate panel, consisting of a collegiate jurisdictional body, composed of first instance judges gathered to act temporarily in the second instance. Although, over constitutional issues, parties can always appeal to the Supreme Court, in cases of questions of federal law, appeals to the Superior Court of Justice are rarely permitted by law.

## Litigiousness and the judiciary

The Brazilian Constitution establishes access to justice as a fundamental right, stipulating, in Article 5, XXXV, that "the law will not exclude from judiciary assessment injury or threats against rights". There is no denying the importance of giving constitutional status to access to justice, especially after two decades of dictatorship, the context in which the 1988 Constitution was promulgated. However, since the Brazilian legal culture has traditionally relied on the judicial process as the instrument for resolving disputes, jurisdiction has been strengthened, without much attention given to either preventive or alternative means of resolving conflicts.

This choice in jurisdiction has resulted in a highly litigious social environment in which citizens are encouraged to approach the judiciary to solve their problems, however simple they may be. As a result, a multitude of lawsuits occur, many of which are repetitive or sometimes even artificially created by opportunistic lawyers for gain. If this situation were not enough, legal proceedings tend to continue for long periods, because of the established dogma that jurisdictional quality can only be assessed in terms of the defense opportunities afforded to the individual, and not by the judiciary's capacity to legitimately resolve disputes.

The model of jurisdiction in Brazil presents several characteristics which increase the volume of legal cases, including appeals. Foremost are characteristics established by the Constitution. Not only does the Constitution define access to justice as a human right and guarantee it. It also contemplates certain directives of social transformation. Consequently, legal cases have become the keystone for control of public policy. Moreover, the Constitution prescribes behaviors over the full spectrum of life and politics. As a result, any matter can be brought before the judiciary.

Also significant are two other facts: First, appealing to higher courts is relatively easy. Second, the civil process has both an individualistic and formalistic profile. This profile makes it difficult to resolve conflicts from a collective perspective, or even to prevent litigation. Accordingly, the legal system discourages consensual conflict resolution and generates a plurality of

individual decisions. This great number of decisions makes it difficult to give precedence its due, further increasing the volume of legal cases.

Access to justice even for the most disadvantaged is also guaranteed by a series of measures that facilitate protection of rights. Going to court is relatively cheap and accessible, even in this country that socially and economically marginalizes a significant part of its population. Court fee waivers are guaranteed to the most needy and in numerous situations, it is possible to litigate even without a lawyer. Citizens count on free services for filing lawsuits even within the judiciary itself, in the small claims courts, and with the Public Defender's Office, whose mission is to provide legal assistance exclusively to those in need.

Brazil has over 1.2 million active lawyers, which creates a large market in search of cases. In this environment of high competition, it is common for lawyers to bargain for fees, making litigation economically attractive for them. In addition to that, the resulting attorney fees make it more profitable for lawyers to litigate than to seek alternative solutions for the litigants. Moreover, law schools train lawyers as experts in litigation, not as litigation analysts.

Historically, Brazilian law does not favor alternative means of resolving conflicts. This characteristic is indicative of another: Brazilian law discourages negotiated solutions, giving little importance to free will. This influences interpretation of legal norms towards a more formal sense, in which jurisprudence is seen in terms of oppositions and conflicts, and not as an instrument for harmonizing interests. And though it is true that, more recently, much has been legislated to change that, such laws are still in the process of assimilation and improvement.

Also, Brazilian law does not traditionally give due value to judicial precedent, weakening the stabilizing function of courts. Only in 2016, the new Civil Procedure Code established binding precedents, a concept that has been improving judicial management of conflicts. However, since most lawyers did not study the topic in law school, the positive effects of the new policy have been felt very slowly.

Finally, a comprehensive defense is often associated with the prerogative of unlimited appeal against judicial decisions. Strictly speaking, Brazilian law allows parties to appeal almost indefinitely, especially in the higher courts. The interesting thing is that our data show a low reversal rate in judicial decisions, which is why it is yet more likely that artificial litigations result from the incentive to appeal.

This combination of factors makes it attractive to choose to resolve conflicts by non-consensual means, in accordance with the traditional model of jurisdiction. Parties more often opt for individual lawsuits. These are prolonged through the excess of both bureaucracy and the lack of available resources. This adds to the number of cases pending trial, which in turn compounds the number of unresolved lawsuits, hindering the capacity of the judiciary to solve cases in a timely manner.

As a result, Brazil has become a highly litigious environment, with 75.4 million lawsuits in progress, a number that has been decreasing in recent years, thanks to the planning and management activities of both the courts and the National Council of Justice, along with massive investment in technology. Although relevant, these activities have not proven sufficient for reducing the backlog of pending cases. The official data presented in the "Justice in Numbers" report from the Brazilian National Council of Justice shows a growing proportion of cases using electronic filing systems, which reached 96.9% of the total number of new cases in 2020. In contrast, the percentage was 11.2% in 2009. These data are particularly important, because they reveal the historical curve of investment in technology, notably in electronic filing systems. Despite these advancements, resolving cases in a timely manner remains a problem in Brazil.

## The federal judiciary

Currently, the federal judiciary is composed of federal judges, who act during the first instance, and five Federal Regional Court, each with jurisdiction over a part of the country. In total, there are 989 federal courts, divided into either general or specialized competence.

Our study used lawsuits that were processed in Federal Small Claims Courts within the jurisdiction of the 5th Regional Federal Court (TRF5, the acronym in the Portuguese language), between 2005 and 2020.

The FSCCs are competent to try cases with a value of up to 60 minimum wages (between US $10,000 and $20,000, according to the historical average of the exchange rate). The main beneficiaries of these cases tend to be the poorest people who seek redress in court for an administrative denial of some social security or assistance benefit in the amount of a minimum wage. The area under the 5th Region jurisdiction has the lowest human development index (HDI) in the country. Accordingly, it is clear that access to these courts in the 5th Region benefits the most needy portion of the Brazilian population.

The jurisdiction of the federal courts has as its main premise, that either the plaintiff or the defendant is the Federal Government, or one of its agencies or enterprises. This means that the federal courts effectively control federal public policy, since in the Brazilian legal system, access to justice against the state is broad and suffers no legal limitations.

On the other hand, compared to the US Constitution, the Brazilian Constitution concentrates the vast majority of public economic and social policy into a much more centralized federal model. As a result, the most expressive portion of federal court activity concerns lawsuits involving social security, healthcare, social assistance, housing, financing, popular banking services, taxation, and educational policy.

The federal government, its agencies and enterprises are represented in court by the Attorney General's office, a giant legal department, with around 7,000 well-paid, highly trained and specialized lawyers, members of the public administration staff. With this structure, it is easy for the federal government to resort to the judiciary as a means to resolve or delay, usually by endless appeals, the resolution of citizens' claims. Thus, the number of appeals in federal courts is enormous, especially those where the government is the appellant.

The principal litigant in the federal courts is the Brazilian Institute of Social Security (INSS, the acronym in the Portuguese language), a federal agency charged with administering social security and social assistance policies. In fact, it is the biggest litigant in the country, even when all branches of the judiciary are considered. For this reason, social security or assistance cases represent about two thirds of all cases tried by FSCCs.

An important aspect about the competence of the federal courts concerns the fact that, in the Brazilian legal system, public policies are generally delineated in the Constitution. Thus, as the federal courts have jurisdictional control involving federal public policies, which are very numerous, judges commonly need to address allegations of constitutional matters in their cases, which facilitates the lodging of appeals to the Federal Supreme Court, which consequently, encourages litigation and further appeals.

## Natural language processing with deep learning

The use of deep learning with Natural Language Processing (NLP) has been proliferating in recent years. This has happened because, since 2018, several researchers have started to use transfer learning methods, improving the quality of models and diminishing the need for large amounts of labeled data [3, 8]. Before that, transfer learning techniques had been uncommon in Natural Language Processing, despite having been very successful in computer vision for several years.

Transfer learning is a technique designed to reuse the knowledge from models between different but similar tasks. Without transfer learning, all model parameters are initialized randomly, and the training process must adjust those parameters to achieve the desired result. With transfer learning, the model parameters are reused from a previous, related task, and the training process will need far fewer data and time to achieve its goal.

We created all classifiers described in this research based on sequential transfer learning, where models learn tasks in a sequence of self-supervised and supervised steps. For a given architecture, we start the process by training a randomly initialized language model with a self-supervised task on general Portuguese text from Wikipedia and from a dataset called "The Brazilian Portuguese Web as Corpus–– BrWaC" [9]. This pretraining phase results in a model that has learned general representations of the Portuguese language. We fine-tune the resulting model using the same self-supervised task on a domain-specific dataset, which in our case is the full text of first instance court decisions.

The specific task used for the language modeling step depends on what was originally proposed by the creators of each architecture. For the ULMFiT method (which uses LSTMs), we trained the model on a Next Word Prediction (NWP) task [3, 10]. For the Transformers-based architectures (BERT and Big Bird), we trained the model using Masked-Language Modeling (MLM) [6, 7]. Finally, we used the encoder of the language model to generate contextual embeddings, which were passed to a classifier. As a result of having reused the linguistic patterns learned during previous steps, the classifier would need far less labeled data than if it had been trained from scratch.

## Long short-term memory networks

In 1997, Hochreiter and colleagues published the first paper on Long Short-Term Memory (LSTM) networks [11]. LSTMs are part of a group of neural networks called Recurrent Neural Networks (RNN). Recurrency allows these networks to keep an internal cell state (memory) at each timestep. The network uses this memory to modify its input signal, making RNNs capable of handling tasks that depend on sequences, like NLP tasks.

As they use the signal from the previous timestep as part of the input for their current calculations, it is easy to understand how errors can build in conventional RNNs. The backpropagation of errors makes simple RNNs hard to train, as they suffer from what is widely known as "vanishing gradient" [12]. Thus, simple RNNs cannot remember long-term dependencies as they cannot learn what to remember and what to forget.

LSTM networks try to solve this problem by learning whether they should remember or forget a piece of information. They can do that due to the presence of three gates that maintain and control the cell state: 1) The input gate regulates which values of the cell will be updated; 2) The output gate controls what parts of the cell it will pass on; and 3) The forget gate controls what information of the cell state it will keep or discard.

Built on LSTMs, the Universal Language Model Fine-tuning (ULMFiT) approach uses a series of tweaks to achieve better results with less training data [3]. In addition to the extensive use of transfer learning steps described above, Howard and Ruder introduced two changes for the optimization steps. The first trick is called "discriminative fine-tuning" and involves using different learning rates for each layer of the network, with the last layer having a higher learning rate than the earlier, deeper ones. The second tweak is called "slanted triangular learning rates" (STLR), a learning rate scheduler. It works by increasing the learning rate (LR) linearly from a minimum to a maximum value during the first 10% of optimizer steps, followed by a linear decay for the rest of the training until it reaches its minimum value again.

## Transformers

Because of the sequential nature of LSTMs, it is challenging to parallelize their computations. Moreover, LSTMs also seem to suffer in handling long-term dependencies, although at a lower degree when compared to vanilla RNNs. To avoid these problems, Vaswani and colleagues [13] replaced recurrency with multi-headed self-attention in an architecture made of an encoder and a decoder. These parts have access to the entire input simultaneously (instead of sequentially, like LSTMs), making Transformer-based models inherently bidirectional. Later, researchers discovered that they could achieve state-of-the-art performance for different NLP tasks using only the encoder part of the original Transformer.

"Bidirectional Encoder Representations from Transformers" (BERT) is the most prominent architecture to use only the encoder part of the Transformer to solve several NLP tasks, like text classification. Each BERT layer is composed of a multi-head attention and a feed-forward sublayer. While we encourage the reader to see the original paper for details [6], the intuition behind the attention mechanism is that all tokens independently attend the entire sequence passed to the model. To represent this idea, one can imagine a matrix of size N x N, where N is the number of tokens fed to the model, and the values inside the matrix indicate the relationship between any given two tokens. That matrix has $N^2$ elements, which means that an increase of the input size by X tokens will require $X^2$ new elements that the model will need to keep in memory and compute. To illustrate, consider that for an input size of 10, the model would need to keep track of 100 elements, whereas a model with 10 times that input size would need 100 times more memory and computational power to accommodate the input. This is why it is said that the attention mechanism has quadratic complexity.

While allowing each token to attend to all others, the attention mechanism of BERT comes with a hefty penalty in terms of memory requirements because of quadratic complexity. This complexity limits the input size BERT can handle (usually 512 tokens), which limits its use with longer texts, including most legal documents. The easiest way to overcome this limitation is to split the text into chunks of 512 tokens, collect the resulting embeddings for each chunk and condense them using some function, as we will show in our experiments section with the BERT + LSTM model. [5]. Despite being relatively easy to implement, this approach results in the loss of long-term dependencies, as the tokens from one chunk cannot attend the tokens from the others.

There are several, more complex, solutions to keep the advantages of the traditional Transformer while allowing them to process longer sequences [14–17]. We used Big Bird [7] to experiment with longer texts. Big Bird uses a sparse attention mechanism to reduce its memory complexity to $O(n)$ instead of $O(n^2)$, but it maintains the properties of the full attention model initially designed for Transformers. The original paper experiments with sequences of up to 4,096 tokens, but we applied several optimization techniques [18] to achieve a model that can handle up to 7,680 tokens.

The training strategy for BERT involves two different self-supervised tasks simultaneously: Masked Language Modeling (MLM) and Next Sentence Prediction (NSP). The training of Big Bird only uses MLM. The MLM task consists of randomly masking input tokens and getting the model to predict their original values. During the NSP task, the model must predict if two sentences are adjacent to each other. The resulting encoder can pass the representation of the input text to various layers, enabling its use for several NLP tasks.

Like the ULMFiT strategy, BERT and Big Bird rely on transfer learning to achieve better results. They are firstly trained on a general domain dataset and fine-tuned on domain-specific text. Then, we pass the embeddings generated by the encoder to a classification layer to achieve our goal.

BERT is the only model with a checkpoint available in Portuguese [19], so we could warm-start from this pretrained model. We trained ULMFiT and Big Bird from scratch using the general Portuguese datasets described above. We used the same effective batch size and learning rates originally proposed by their authors. Next, we fine-tuned all models using first instance court decisions as our domain-specific texts. At the end of this process, we obtained language models from the three architectures that understood legal text written in Portuguese.

## Methodology

Considering this critical role played by FSCCs, we selected all of them within the 5th Regional Federal Court, which has jurisdiction over six Brazilian states that are home to 15.13% of the country's population: Alagoas (AL), Ceará (CE), Pernambuco (PE), Paraíba (PB), Rio Grande do Norte (RN) and Sergipe (SE). In addition to the Regional Court itself, there are 127 sub-courts with 10 appeal sectors. Currently, there are 228 judges, of whom 15 are appellate judges. Considering the workforce, it is the smallest Federal Regional Court in Brazil. However, the workload is the highest, with an average annual productivity of 3,182 cases tried per judge in general courts and 9,267 cases tried by each judge in the FSCCs.

Currently, all proceedings in progress at the 5th Regional Federal Court are electronic, using one of two systems: "PJe" in general jurisdiction courts, or "CRETA", used in FSCCs since 2004 and being gradually replaced by PJe since 2021. The sample used in our research is restricted to the CRETA system, because it contains the history of all cases tried by FSCCs between 2004 and 2020.

We collected data from their public repositories, totaling 3,128,292 lawsuits filed between June 2004 and February 2020, of which we managed to clean and label 765,602 appeals tried over the same period. To our knowledge, this is the most extensive study of its kind in Brazil.

To handle this kind of data, we have selected only architectures that can handle longer texts, as we commonly find in judicial decisions, where judges do not have any limitation regarding the number of words they can use. All models use only text as their inputs, specifically, the full-text content of the final trial court ruling, with no preprocessing other than model-specific text splitting described in each model section and the removal of web scraping artifacts, like headers and HTML tags. All texts are in Portuguese, the official language of Brazil.

We also asked 22 experts (5 federal judges and 17 federal judicial clerks) to predict outcomes for 690 appeals randomly sampled from our test dataset. Each appeal in this dataset was analyzed by one person only, as we will further explain in the "Legal experts' analysis" section under "Methodology". The experts had an average of 11 years of previous experience working within federal courts, of which 8 years were working exclusively on FSCCs. We sampled experts opportunistically, inviting judges and clerks who we were sure to have previous experience with FSCCs. We called judges individually and invited them and their clerks to participate in the research. After the initial contact, we sent instructions so they could register and use the data annotation system [20], which was available only to people with IP addresses from the court. We admitted all participants who indicated an interest in our invitation, and they were free to predict the outcomes for as many cases as they wanted to.

Pursuant to the Brazilian National Council of Health Resolution n. 510/2016, article 1, item VII, researchers in humanities and social sciences (including law) are not required to submit their research for REB review when the research does not involve the collection of personal identification of participants or when the researchers do not publish personally identifiable information and the information collected is related to subjects' professional practice. For this reason, we could not collect demographic details about participants other than the information below.

We asked participants to register through a website (a data labeling system) where they did not have to inform their name or other personable identifiable information other than: 1) if they were a Judge or a Clerk; 2) how many years of experience they had within the Judiciary and 3) how many years of legal education they had. This website was designed in a way so we could avoid external participation.

Participants assessed a legal decision and predicted the outcome of an appeal for that case. That is part of their acceptable standard professional activities. Civil servants in the Judiciary are routinely requested to provide these opinions to individuals and organizations.

Furthermore, we obtained all datasets from publicly available systems, which also exempts that part of our research of REB review (Brazilian National Council of Health Resolution n. 510/2016, article 1, item II). The Brazilian legislation does not require us to submit our research REB review for the reasons above.

## Our dataset

We collected data publicly available for all electronically filed cases within all FSCC under the 5th Regional Federal Court jurisdiction. According to the Brazilian constitution (article 5, LX, and article 93, IX), court records are considered public and must be publicly available for inspection by anyone. Only in exceptional cases can records be sealed due to their nature or by a judge's decision. FSCC lawsuits are almost always public, being easily accessible by anyone with an internet connection. While full access depends on providing user credentials, basic case information is freely available, including plaintiffs' and defendants' names, case details, and judicial opinions.

Starting in 2004, FSCCs in the 5th Regional Federal Court jurisdiction began to use an electronic filing system, and, in 2006, all new cases and appeals became 100% electronic. Therefore, our initial dataset consists of 3,128,292 lawsuits filed between September 2004 and February 2020. Due to the legal costs associated with an appeal, most disputes do not get past the lower court, and an AP hears roughly 25% of all cases.

Considering our proposed task, we narrowed our initial dataset to contain only records where an AP actually heard an appeal. After some data cleaning, we ended up with a working dataset containing 765,602 appealed lawsuits. We release the dataset with this paper, naming it the "Brazilian Courts Appeals Dataset for the 5th Regional Federal Court—BrCAD-5" along with its datasheet [21]. It includes the case number, basic case metadata, the full text of both the first instance court and the appellate panel decisions, and the target label for our task (see the data in S1 Appendix for a complete datasheet).

To keep it simple, we labeled as "reverse" any case where the AP accepted at least one appellant's arguments and reversed the lower court decision, whereas all others were labeled as "affirm". This approach is useful for computational reasons, as we simplify our analysis by converting a multiclass classification problem into a binary one. It is also legally sound, as both partial and full reversed appeals exempt the plaintiffs in error to pay legal expenses, which are due when they lose the appeal (that is, when the AP decides to "affirm" the FSCC ruling). Table 1 displays the distribution of Reverse and Affirm for our entire dataset:

**Table 1. Outcomes distribution on our dataset.**

| Class | Number of events |
|---|---|
| Reverse | 111,276 (full reverse) + 49,259 (partial reverse) = 160,535 |
| Affirm | 605,067 |
| Total | 765,602 |

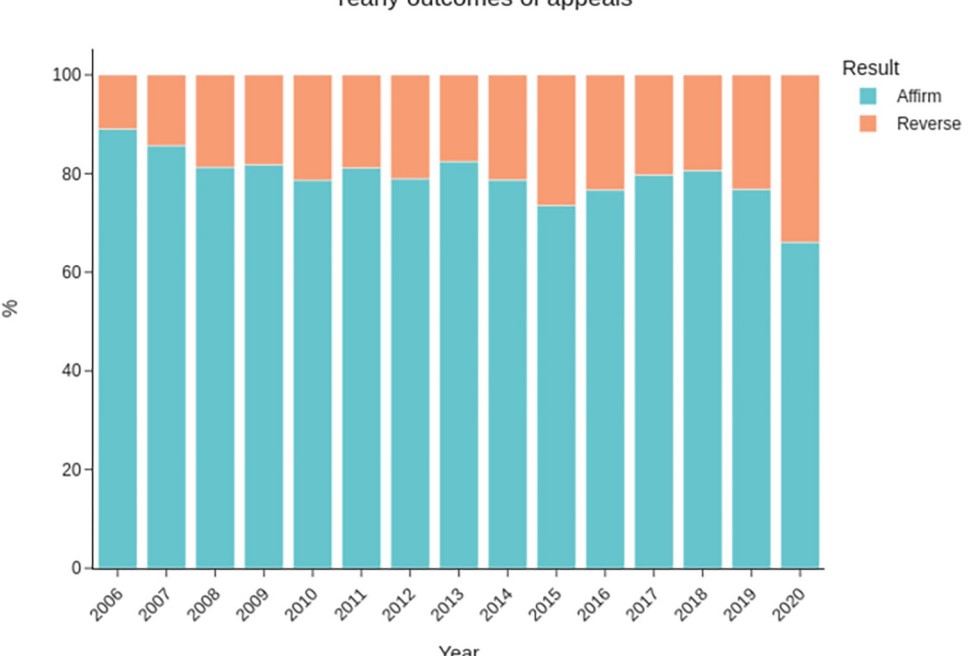

**Fig 1. Yearly distributions of results of appeals against rulings from Federal Small Claims Courts within the jurisdiction of the 5th Regional Federal Court.** The data for 2020 contains appeals tried during the first trimester.

As shown above, appellate panels affirm lower court decisions in 79% of cases. As a consequence of losing, most appellants have to pay legal expenses for the prevailing party, representing a significant economic impact for those seeking to modify previous rulings. Fig 1 demonstrates the stability of these proportions over time.

## Data preparation

Gathering correct labels for our classifier was not easy. In addition to the full-text of first instance decisions, we had to collect our target variable, that is, the outcome of appeals. This information may be present within the case metadata, but it was not always available, and, sometimes, the labels were just obviously wrong. Such errors happen when court clerks mishandle cases within the electronic system or when they fail to rename the file uploaded to the system.

We used several heuristics to identify and assess the quality of our labels from three data points: 1) case metadata; 2) filename containing the AP ruling; 3) hand-crafted regular expressions to extract the label from text of the AP ruling. While creating these heuristics, we sampled the data and verified labels manually several times to ensure their quality. In the end, we used a simple voting classifier in which at least two out of three label sources must agree in order to consider that label valid.

One important point to note is the method by which we split train/validation/test datasets. For those unfamiliar with this naming scheme: In simple terms, we use the training dataset to train our model, the validation dataset to adjust hyperparameters/architectures, and the test dataset to present our final results. The most common way to split the data is the holdout method, where researchers keep one randomly chosen part of their dataset (usually 20%) as validation and test data.

While some researchers facing similar LJP tasks choose this way to split their data [22, 23], there is a critical methodological flaw when they randomly sample cases as their validation dataset. Traditional random splitting not only could but has been proven to generate overly optimistic results that would not be achievable in production [22]. Other models [1], while not falling into the same problem, also depend on partial knowledge about the outcome that would not be available during production. Katz and colleagues [24] named these traps "out of sample applicability", which is, according to them, one of the three principles that ML approaches for quantitative legal prediction must follow to be deemed useful.

This problem happens because neither random splits nor K-fold cross-validation consider the time-sensitive nature of legal decisions. That is, the law changes over time, and so do courts and their decisions. Given any two similar cases, they are much more likely to have the same fate if they are decided within a closer timeframe. That could pose a relevant data leakage problem, where information from the future leaks into the training dataset, granting access to data that the model would not have in production [25].

To avoid such pitfalls, we arranged the data sequentially in time (considering the date of the appellate panel ruling) and selected the first 80% of entries as training data, while randomly splitting the last 20% into two halves of validation and test data. This way, we ensured no information from the future would leak into the model during training. The training dataset contains 612,961 appeals tried between January 23, 2006, and March 26, 2018. The validation and the test datasets have, respectively, 76,342 and 76,299 entries, ranging from March 26, 2018 to April 01, 2020. Fig 2 illustrates the issue with constructing training, validation, and test datasets and how we dealt with it.

## Metrics strategy

Researchers working with LJP tend to use accuracy or F1-score to explain the performance of their models. Nonetheless, both metrics are insufficient to describe how well a model would

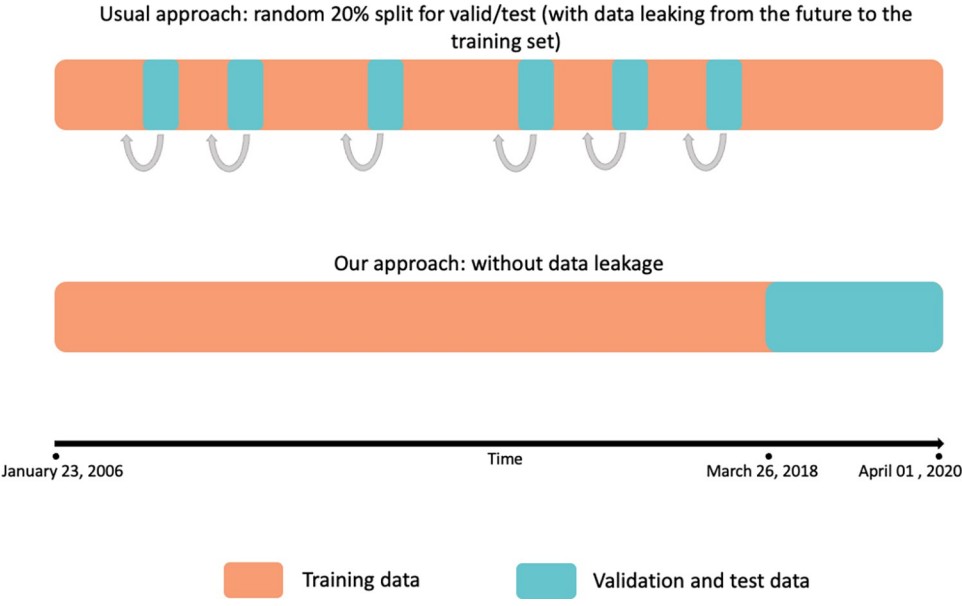

**Fig 2. Comparison between traditional holdout method and our time-sensitive approach.** Grey arrows show how, in the traditional holdout method, data leaks from the training set into the validation and test sets, which leads to overly optimistic results, as the model has access to information from the future.

perform in the real world, given that most appeals affirm the previous ruling. As explained by Chicco and Jurman [26], accuracy and F1-score can show overly optimistic results, especially on unbalanced datasets.

Following their work and others [27, 28], we have decided to use the Matthews Correlation Coefficient (MCC) as the primary way to evaluate our classifiers. The MCC measures the differences between expected and predicted values, being similar to the chi-square statistic for a 2x2 contingency table. For a binary problem, we have $MCC = \frac{TP \times TN - FP \times FN}{\sqrt{(TP+FP) \times (TP+FN) \times (TN+FP) \times (TN+FN)}}$, where TP, TN, FP and FN are, respectively, true positives, true negatives, false positives, and false negatives.

Unlike accuracy and F1-score, MCC considers all elements of the confusion matrix (TN, TP, FP, FN), providing a better view of the performance of classifiers. It is also easy to interpret, with values ranging from -1 (worst case) to +1 (best case) and 0 considered as good as random guesses.

## Experimental setup

We have tested three different deep learning architectures. In all cases, we posed the problem as a binary classification task. The classifier input was only the full text of the first decision, which is available at the end of the trial and before the moment when the losing parties must decide if they will appeal. By using only this text, we ensured our model had no more information than would be available to all parties. We made the links for all code, datasets and model binaries available in S1 Appendix.

Unless otherwise specified, for the language modeling part, we used the hyperparameters described in the original publications of the architectures. For the classification task, we used Bayesian optimization for hyperparameter tuning [29, 30]. For each architecture, we provide in the following tables the hyperparameters used to train our final classifiers.

The first classifier follows the ULMFiT method described by Howard and Ruder [3]. Using their approach, which includes their hyperparameters, we trained two language models, one of which reads the text from left to right and the other from right to left, using the Portuguese Wikipedia dump. After that, we used all 3,128,292 first instance court decisions, including those without an appeal, to fine-tune the language model on legal text. Finally, we trained our classifier on top of that language model.

The training of the ULMFiT classifiers (backward and forward) followed the original publication of the methdology, with a few tweaks. We adjusted the intermediate hidden size to 100 (instead of 50) and the dropout probability to 0.3 (instead of 0.1). The maximum learning rate was 0.001. We obtained the maximum learning rate using the FastAI [31] implementation of the "LR range test" algorithm [32]. We trained the final classifier for 20 epochs using the One Cycle Policy [33]. For the first four epochs, we froze the LSTM encoder, and only the classifier layer had its parameters updated with the maximum learning rate. Following this, we applied discriminative fine-tuning and gradual unfreezing of the encoder layers for another 16 epochs. Table 2 shows the hyperparameters we used for this model.

We wanted to compare the ULMFiT classifiers with other approaches, mainly with recent advancements made using the Transformer architecture [6] which achieves state-of-the-art results in several natural language processing tasks. There are, however, several limitations on sequence length due to the memory cost of the attention mechanism of this new architecture. While a few approaches try to overcome this obstacle [7, 14, 15], popular Transformers maintain this sequence length constraint. In our case, this limitation poses a problem, considering the widely variable lengths of our input texts. The decisions in our dataset can have up to

**Table 2. Hyperparameters for the ULMFiT classifiers.**

| Hyperparameter | Value |
|---|---|
| Effective batch size | 64 |
| Maximum learning rate | 0.001 |
| Optimizer | Adam [34] with betas = (0.8, 0.7) and weight decay = 0.01 |
| Learning rate scheduler | One Cycle [33] |
| Epochs | 20 |
| Dropout | 0.3 |
| Output sizes of the classifier layer | 100, 2 |
| Activation function of the classifier layer | ReLU [35] |
| Loss function | Binary CrossEntropy |

20,000 words per document, with 86% of them being longer than the 512 tokens supported by most Transformers.

To overcome such restrictions, we designed the second classifier following the work of Mulyar and colleagues [5]. We split each document containing the court ruling into chunks of 512 tokens. After that, we fed them to a Portuguese BERT [19] to gather all $\frac{text\ length}{512}$ hidden states of the CLS token from the last four layers. The CLS token is the aggregate representation for that entire input. Fig 3 illustrates an overview of the process for this model.

We used one unidirectional LSTM layer with an output size of 1,536 to condense the language model hidden state sequences into one tensor containing the final output of the LSTM, which we pass through a final linear layer for classification. This last classification part has two linear layers with output sizes of 384 and 1, combined with dropout with p = 0.1 and using Mish [36] as the activation function. During the training of the classifier, we froze all but the last layer of the BERT encoder. Finally, we applied a sigmoid function with a threshold of 0.49 to the output of our last layer to obtain our final classification. Table 3 shows the hyperparameters we have used for this model.

For the third classifier, we decided to follow recent trends on the use of linear complexity transformer models. To achieve this, we trained one Big Bird [7] model from scratch. We designed it to handle texts up to 7,680 tokens long using both the Portuguese Wikipedia and the BrWaC dataset [9] as our initial corpus. We applied the Zero Redundancy Optimizer [18] to allow this sequence length without facing memory constraints.

We trained this language model with Portuguese text for 142,800 steps using an effective batch size of 256 and achieving a training loss of 1.6346. After that, we fine-tuned this language model on the same dataset we used for ULMFiT before, containing all 3,128,292 first instance court decisions for one epoch.

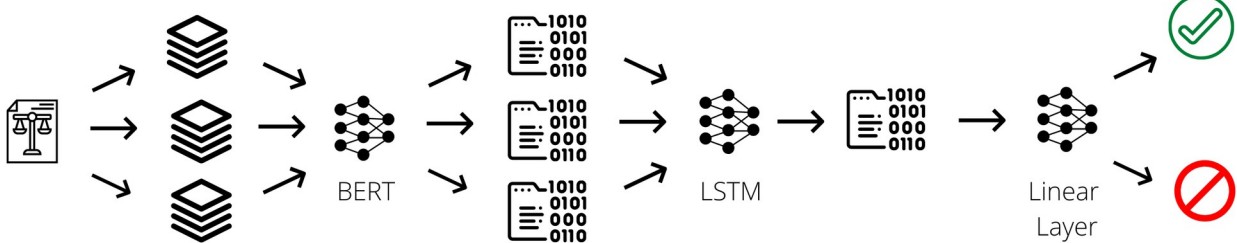

**Fig 3. Overview of the BERT + LSTM model.** The text of a court ruling is split into chunks containing 512 tokens each. They are passed to a Portuguese BERT model, from which we collect the embeddings from the CLS token. We feed an LSTM with the embeddings from the previous step, condensing them into one vector. We pass this vector to a classifier head to get a final classification.

**Table 3. Hyperparameters for the BERT + LSTM classifier.**

| Hyperparameter | Value |
|---|---|
| Effective batch size | 48 |
| BERT batch size | 15 |
| Learning rate for the classifier | 0.0003 |
| Learning rate for the encoder | 0.00002 (only the last BERT layer was not frozen) |
| Optimizer | AdamW [37] with betas = (0.9, 0.999) and weight decay = 0.1 |
| Learning rate scheduler | One Cycle [33] |
| Epochs | 8 |
| Dropout | 0.1 |
| Activation function of the classifier layer | Mish [36] |
| Loss function | Binary CrossEntropy |

In a similar fashion to what we did with the previous architecture, we used this Big Bird model to generate features, but without the need to use an LSTM to condense the hidden states from the encoder. Therefore, we concatenated the hidden states from the last four layers of the encoder for the CLS token and passed it to classification layer. This last classification part has three linear layers with output sizes of 768, 384, and 1, intercalated with dropout with p = 0.3, and using Mish [36] as the activation function. During the training of the classifier, we kept all but the last layer of the Big Bird encoder frozen. Finally, we applied a sigmoid function with a threshold of 0.49 to the output of our last layer to obtain our final classification.

When training the classifier, if our input text was longer than 7,680 tokens, we split the text to use the first 3,840 and the last 3,840 tokens, omitting the text in between. This is one limitation for texts longer than 7,680 tokens, but, in our legal experience, the beginning and the end of a court decision retain the information needed to understand the case under trial. Table 4 shows the hyperparameters we have used for this model.

## Legal experts' analysis

It would be nearly impossible for our expert panel to evaluate all entries within our validation and test datasets. Therefore, we randomly sampled 690 cases from it, resulting in a dataset highly representative of the original data, as seen on the Table 5:

We then asked 22 highly skilled professionals to predict the outcome for the appeal of the sampled cases. Five experts were federal judges, and the other 17 were federal judicial clerks who provide everyday assistance for federal judges. All participants held a law degree and had previous experience working in FSCCs. For their task, they had to use only the same data

**Table 4. Hyperparameters for the Big Bird classifier.**

| Hyperparameter | Value |
|---|---|
| Effective batch size | 32 |
| Learning rate for the classifier | 0.0003 |
| Learning rate for the encoder | 0.0003 |
| Optimizer | AdamW [37] with betas = (0.9, 0.999) and weight decay = 0.1 |
| Learning rate scheduler | One Cycle [33] |
| Epochs | 15 |
| Dropout | 0.3 |
| Activation function of the classifier layer | Mish [36] |
| Loss function | Binary CrossEntropy |

**Table 5. Size and label distribution for the validation, test and human experts' datasets.**

| Dataset | Number of entries | Proportion of appeals affirming the decision from first instance court | Proportion of appeals reversing the decision from first instance court |
|---|---|---|---|
| Validation | 76,342 | 0.221136 | 0.778864 |
| Test | 76,299 | 0.221261 | 0.778739 |
| Human legal experts | 690 | 0.210145 | 0.789855 |

available for our models: the full-text of the first instance court decision. Table 6 contains some relevant information about our expert group. We weighted all values by the number of cases each participant labeled.

Using an open-source annotation tool [20], experts were free to evaluate as many appeals as they wanted to, which led to values ranging from 5 to 56 labels per participant (μ = 31.36, σ = 14.67). Each appeal was analyzed by one expert only. We compare their performance against our models in the section "Results" below.

## Results

After assessing several hyperparameters on our validation dataset, we selected the set of hyper-parameters that provided the best value of MCC. We provide our results on the test dataset and on the human-evaluated dataset. Table 7 shows the MCC for each classifier.

As seen above, all models outperformed the highest estimation of the human expert group (MCC = 0.226), considering the 0.99 confidence interval. With the best MCC, the ULMFiT classifier shows that there is still room for using RNNs, especially when considering the memory limitations of the Transformers.

Assuming the always-changing nature of the law, we were expecting to see some drift while analyzing the performance of our model on the test dataset. In simple terms, data drift occurs when statistical properties of our data continuously change, decreasing the performance of the model over time. Such a drift can happen suddenly in law, for instance, when Congress approves a new bill, changing how courts interpret laws and their connections within the legal system. However, data drifts can also happen slowly, as the interpretation of laws usually changes over time, even without new legislation.

We have found no drift or decrease in the capacity of our models over the timeframe of our test dataset containing cases tried by appellate panels between March 2018 and April 2020. On the contrary, as seen in Fig 4, the monthly MCC for the predictions remained relatively stable over time, even showing an increase after October 2020, with an MCC of 0.56 considering appeals tried in February 2020.

## Limitations

At this early phase, our model only analyzes the text of decisions from first instance courts in order to predict whether or not a given decision will be overruled by the appellate panel. Hence, we presuppose, for now, that each first instance ruling contains the signal needed to

**Table 6. Experience and education data about the experts that labelled our dataset.**

| Metric | Weighted average time (in years) |
|---|---|
| Legal higher education | 6.41 |
| Experience in Federal Courts (including Small Claims Courts) | 11.21 |
| Experience in Federal Small Claims Courts exclusively | 8.02 |

**Table 7. Performance results on the human experts dataset and test dataset.** We obtained the 0.99 confidence interval based on the Fisher r-to-z transformation.

| Architecture | MCC on the test dataset | MCC on the human-experts dataset / 0.99 CI |
|---|---|---|
| Human experts | - | 0.1253 / (0.022–0.226) |
| ULMFiT forward | 0.3238 | 0.2768 / (0.178–0.37) |
| ULMFiT backward | 0.3544 | 0.3531 / (0.259–0.441) |
| ULMFiT bidirectional | **0.3688** | 0.3367 / (0.241–0.425) |
| BERT + LSTM | 0.3127 | 0.3326 / (0.237–0.422) |
| Big Bird | 0.2649 | 0.1857 / (0.089–0.279) |

predict the outcome of an appeal. For this reason, we disregard other potential sources of relevant information, like the arguments raised by the parties in their appeals and rebuttals.

A legal scholar might note that the result of an appeal will also depend partly on the arguments brought by lawyers during trial. These arguments are completely ignored by our models, since we do not use these texts as inputs. Future works may use these arguments as well as texts from other sources in order to extract the signal leading to the appellate panel decision.

## Discussion

LJP researchers usually focus on higher courts from wealthy countries, like the European Court of Human Rights [22] and the U.S. Supreme Court [1, 24, 38, 39], although using very diverse approaches. In some instances, researchers used lower court data to distinguish confusing law articles [40], classify a case's current status in the legal system, or predict charges, applicable law articles, and prison terms [41]. In particular, one study tried to predict appeal outcomes in a Brazilian state court, which is very similar to our goal [23].

Most studies rely on relatively small datasets, using only a few hundred or thousand examples. The exceptions are Chinese studies, which use a publicly available dataset called Chinese AI and Law Challenge (CAIL2018) with around 1.2 million cases. Sometimes, researchers train models using raw text data, while others use case metadata and rely heavily on feature engineering.

Like previous works on LJP, we chose a metric and aimed to optimize it. However, it is crucial to note that most studies fail to show how useful a model really is. While there is nothing inherently wrong with this approach (as long as they take these limitations into account), we understand that it is not enough to assess the overall performance of ML models using solely traditional metrics. Despite being easy to calculate, such metrics cannot evaluate, on their own, the value and usefulness of such models in real case scenarios.

The usefulness of previous studies in this area is usually unclear, mainly due to low-quality baseline models used for comparison. Some authors [24] suggest using simple heuristics, like "always guess reverse" or slightly more sophisticated approaches, such as employing a variable-length moving window to calculate the most common result during that period. In any case, we are sure that no legal expert would deem such baseline models useful, as they would not add any intelligence to their analysis. It is quite easy to get results that are marginally better than random guesses, but that is not what judges and lawyers have in their minds when they talk about LJP.

Instead, we used human experts as our baseline model to compare against machine learning. This is not entirely new, as other researchers [38] have already performed this kind of comparison, using human experts as the baseline for an LJP model. Like those authors, we also can provide a better insight into the real value of an ML solution for our proposed task.

A computer-based approach to predict legal cases is only useful if it achieves human-like performance, that is, if it can provide at least the same quality as human analysis for a fraction

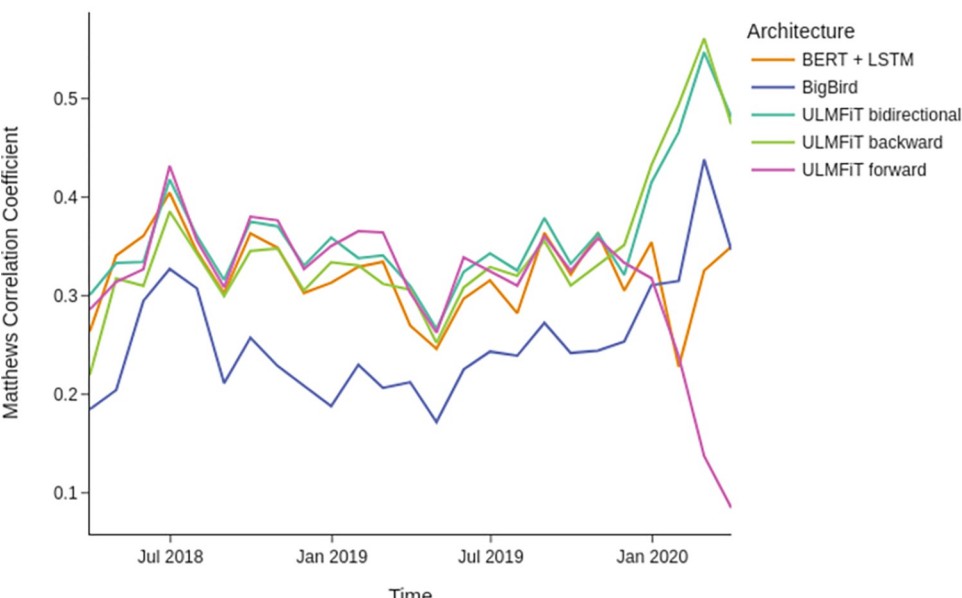

**Fig 4. Matthews Correlation Coefficient for each architecture on the test dataset.** We calculated the MCC considering all appeals tried each month. The monthly MCC for all models remains constant during the timeframe of the test dataset, showing no evidence of data drift.

of the time and cost. That is because, as we believe, it is reasonable to argue that highly skilled professionals represent the highest cost of most legal businesses and institutions.

Our results show that we were able to train a model that can predict the outcomes of appeals in cases within the Brazilian 5th Regional Federal Court better than human experts with a solid legal background. Based on this result, we propose that our tool is helpful from a practical point of view. It can help manage knowledge associated with judicial activity, which is the primary goal of intelligence centers. Although we have only used data from the 5th Region, we are confident that the same logic would apply to all federal courts in Brazil. Our models provide an extra layer of analysis that can be useful for both judges and the legal system itself. Judges usually need to rely on their own memory and do extensive research on legal precedents to try a case. As a result, by using our tool, the legal system could achieve better stability and treat cases more consistently. It is not enough to remember that stability and consistency are main reasons the legal system exists.

In addition to that, our model could be helpful as a tool to provide better information for lawyers as they decide whether or not to appeal, avoiding unnecessary prolongation of lawsuits and the payment of legal fees. Furthermore, the legal system would also benefit from a decrease in the number of pending cases, which would enable the judiciary to focus its resources on more pressing matters. We hope that, by making additional analytics available to all parties involved in trials, they will be able to make more informed decisions. This will have a positive effect on the role they play within a more efficient legal system.

## Conclusion

We have shown that it is possible to use deep learning models to predict outcomes of appeals in Brazilian courts, achieving performance that is better than that resulting from analysis by human experts. We found that the ULMFiT methodology (AWD-LSTM) performed better for

this specific task, showing that there is still room for Recurrent Neural Networks, especially when considering longer texts, like judicial decisions.

We also provide a new dataset called the Brazilian Courts Appeals Dataset for the 5th Regional Federal Court—BrCAD-5 which can be used in the future to establish a common ground for this kind of modeling and foster future endeavors in LJP tasks within the Brazilian judiciary.

## Supporting information

**S1 Appendix. Code, datasets and datasheet for the Brazilian Courts Appeals Dataset for the 5th Regional Federal Court—BrCAD-5.**
(DOCX)

## Acknowledgments

We thank the following people for their invaluable help labeling our expert dataset: **C.D. Fonseca**; **L. C. F. A. Aquino**; **E. S. Medeiros**; **I. S. R. A. Dantas**; **M. S. L. C. Balliana**; **A. C. D. Ferreira** and their peers at the 5th Regional Federal Court.

We also thank M. C. M. Jacob for her helpful comments and suggestions, Teacher Jonathan for his help in polishing our writing, and the High-Performance Computing Center at UFRN (NPAD/UFRN) for the use of their GPU infrastructure.

## Author Contributions

**Conceptualization:** Elias Jacob de Menezes-Neto, Marco Bruno Miranda Clementino.

**Data curation:** Elias Jacob de Menezes-Neto.

**Formal analysis:** Elias Jacob de Menezes-Neto.

**Funding acquisition:** Elias Jacob de Menezes-Neto.

**Investigation:** Elias Jacob de Menezes-Neto, Marco Bruno Miranda Clementino.

**Methodology:** Elias Jacob de Menezes-Neto, Marco Bruno Miranda Clementino.

**Project administration:** Elias Jacob de Menezes-Neto.

**Software:** Elias Jacob de Menezes-Neto.

**Validation:** Elias Jacob de Menezes-Neto.

**Visualization:** Elias Jacob de Menezes-Neto.

**Writing – original draft:** Elias Jacob de Menezes-Neto, Marco Bruno Miranda Clementino.

**Writing – review & editing:** Elias Jacob de Menezes-Neto, Marco Bruno Miranda Clementino.

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
