## [Decision Letter · Decision Letter 0]

4 Jul 2022

PONE-D-22-07895Using deep learning to predict outcomes of legal appeals better than human experts: a study with data from Brazilian federal courtsPLOS ONE

Dear Dr. Jacob de Menezes-Neto,

Thank you for submitting your manuscript to PLOS ONE. After careful consideration, we feel that it has merit but does not fully meet PLOS ONE’s publication criteria as it currently stands. Therefore, we invite you to submit a revised version of the manuscript that addresses the points raised during the review process.

 Please address all the recommendations by the two reviewers.

We look forward to receiving your revised manuscript.

Kind regards,

Donrich Thaldar

Academic Editor

PLOS ONE

Journal Requirements:

2. In order to improve reporting, in your methods section, please provide additional information about the participant recruitment method and the demographic details of your participants.

This study was funded by the National Council for Scientific and Technological Development (CNPq) through a scholarship to Elias Jacob de Menezes-Neto (302668/2020-9).

The Brazilian Coordenação de Aperfeiçoamento de Pessoal de Nível Superior (CAPES) financed the fee to publish this article (finance code XXX). 

Funders had no role in the study design, data collection and analysis, or the decision to prepare and publish the manuscript.

This study was funded by the National Council for Scientific and Technological Development (CNPq) through a scholarship to EJDMN (302668/2020-9).

I have read the journal's policy and the authors of this manuscript have the following competing interests:

Elias Jacob de Menezes-Neto declares no competing interests.

Marco Bruno Miranda Clementino is a federal judge in the 5th Regional Federal Court jurisdiction. Although his position is not affected in any way by this paper, we understand that this affiliation may be seen as a non-financial competing interest. 

5. Please remove your figures from within your manuscript file, leaving only the individual TIFF/EPS image files, uploaded separately.  These will be automatically included in the reviewers’ PDF.

Reviewers' comments:

Reviewer's Responses to Questions

**Comments to the Author**

1. Is the manuscript technically sound, and do the data support the conclusions?

Reviewer #1: Partly

Reviewer #2: Yes

2. Has the statistical analysis been performed appropriately and rigorously? 

Reviewer #1: Yes

Reviewer #2: Yes

3. Have the authors made all data underlying the findings in their manuscript fully available?

Reviewer #1: Yes

Reviewer #2: Yes

4. Is the manuscript presented in an intelligible fashion and written in standard English?

Reviewer #1: Yes

Reviewer #2: Yes

5. Review Comments to the Author

Reviewer #1: * Abstract

- Sentence 3: This sentence does not motivate the need for the study as this

study does not really help understand the steps of case flow.

- Line 8: machines  models

- Line 7 - 9: We compare the predictive performance of the models to the

predictions of twenty two highly skilled legal experts.

- Line 10: ... of 0.3688 compared to 0.1253 from the human experts.

- 2nd last sentence - Rewrite the conclusion to be more specific: Our results

demonstrate that natural language machine learning techniques is a promising

approach for predicting legal outcomes.

** Introduction

Page 3:

- Line 3 : admitted  submitted

- First paragraph: "We propose the ..."

- Should specifically mention that the class of models used in this paper are

deep learning, natural language processing (NLP) models.

- Last line: analysis conducted  predictions determined ...

- Second paragraph: "we evaluated ..."

- Line 5: delete "we used"

- Line 7: replace the first "architecture"  "model"

- Third paragraph: "We discuss how ... "

- Line 1: "an algorithm"  "models"

- Sentence 1: Begin the paragraph with: AI-driven systems could have

real potential to improve stability, predictability ... Adust the

paragraph appropriately.

- While the benefits of such systems are described - it must be mentioned

that much more work is required before automated systems replace human

decision making.

** Institutional background

- This section must be reduced - while it may be usefull background - much of it

is not necessary to understand the rest of the paper. It should rather be

reduced from 5 pages to 2 or so pages.

- I would rather see the space used to give a background to the NLP models

used in the study.

** NLP Models Section

- The paper should include a background on the NLP models used in the study. For

e.g. LSTM and Transformer models should be described.

** Methodology

- Paragraph Two:

- clarify if any pre-processing was done - either no preprocessing, or if any

then what did this entail.

- Paragraph Three:

- did each participant predict each of the 690 appeals?

- In what language are the full text written in?

- Page 13 - first two paragraphs - would better fit in the introduction that

discusses the motivation for the study.

- Page 14:

- While the labelling process is sensible, it is not fool proof. Where

there any processes used to verify the labels? For e.g. sampling some of

the decisions?

- Page 16:

- First paragraph in "Experimental setup"

- ".. to all parties in production. "  "... to all parties."

- Experimental Setup:

- The training set up of the different models should give the hyper

parameters used and explain how these were tuned.

- Some more details of the model set up would be useful if they differed

from the standard set up discussed in the literature.

- Legal expert's analysis

- "... 17 highly skilled ..." This number contradicts the numbers given

earlier in the paper - where 22 is reported. Please fix.

- Page 19:

- While we understand the difficulty to obtain human expert decisions,

it would have good to evaluate inter-human agreement i.e. if

participants were to have evaluated the same cases.

** Results

- The hyperparameters must be clearly specified in the methods.

** Discussion

- Paragraph Two:

- Line 2: "example The"  "examople. The"

- Paragraph Four:

- Sentence Two is not substantiated: How will the tool help to detect

inconsistencies - as the tool does not, as described, cluster cases on

similarity. So this entire paragraph is problematic and should be left out

or substantiated carefully.

Reviewer #2: The paper provides an interesting contribution to the literature.

I believe that the paper could improve the explanation of the models that are used and provide the codes so other researchers can follow these steps.

It could open an important reseach area applied to judicial sentences and increase the potential for citations of the paper.

6. PLOS authors have the option to publish the peer review history of their article (what does this mean?). If published, this will include your full peer review and any attached files.

Reviewer #1: No

Reviewer #2: No

---

## [Author Response · Author response to Decision Letter 0]

13 Jul 2022

Natal/Brazil, July 13th, 2022

Dear Editor,

We appreciate the careful reading of our manuscript. We have prepared a revised version of the paper that takes into account all of the comments made by the academic editor and reviewers. Below, we reply to the comments, specifying the text changes. 

We look forward to receiving your decision.

Sincerely, 

Elias Jacob de Menezes-Neto

(on behalf of the authors)

A) JOURNAL REQUIREMENTS

Done.

2. In order to improve reporting, in your methods section, please provide additional information about the participant recruitment method and the demographic details of your participants. 

Done. Please see lines 374-378.

Pursuant to the Brazilian National Council of Health Resolution n. 510/2016, article 1, item VII, researchers in humanities and social sciences (including law) are not required to submit their research for REB review when the research does not involve the collection of personal identification of participants or when the researchers do not publish personally identifiable information and the information collected is related to subjects’ professional practice. For this reason, we could not collect demographic details about participants other than the information already in the manuscript.

3. Please remove any funding-related text from the manuscript and let us know how you would like to update your Funding Statement.

We removed the funding statement from the manuscript and amended the cover letter to update our funding statement.

Done

5. Remove figures from the Manuscript.

Done.

6. Include captions for the Supporting Information at the end of the manuscript.

Done.

7. Review reference list

We had to include several new items on the reference list to accommodate the requests made by the reviewer #1 to provide more details about our models. These were added:

Peters ME, Neumann M, Iyyer M, Gardner M, Clark C, Lee K, et al. Deep contextualized word representations. arXiv; 2018. Available: http://arxiv.org/abs/1802.05365

Soam M, Thakur S. Next Word Prediction Using Deep Learning: A Comparative Study. 2022 12th International Conference on Cloud Computing, Data Science & Engineering (Confluence). 2022. pp. 653–658. doi:10.1109/Confluence52989.2022.9734151

Hochreiter S, Schmidhuber J. Long Short-Term Memory. Neural Computation. 1997;9: 1735–1780. doi:10.1162/neco.1997.9.8.1735

Hochreiter S. The Vanishing Gradient Problem During Learning Recurrent Neural Nets and Problem Solutions. Int J Unc Fuzz Knowl Based Syst. 1998;06: 107–116. doi:10.1142/S0218488598000094

Vaswani A, Shazeer N, Parmar N, Uszkoreit J, Jones L, Gomez AN, et al. Attention is all you need. Advances in Neural Information Processing Systems. 2017. pp. 5999–6009. 

Zhou H, Zhang S, Peng J, Zhang S, Li J, Xiong H, et al. Informer: Beyond Efficient Transformer for Long Sequence Time-Series Forecasting. arXiv; 2020. doi:10.48550/ARXIV.2012.07436

Pappagari R, Zelasko P, Villalba J, Carmiel Y, Dehak N. Hierarchical Transformers for Long Document Classification. 2019 IEEE Automatic Speech Recognition and Understanding Workshop, ASRU 2019 - Proceedings. 2019; 838–844. doi:10.1109/ASRU46091.2019.9003958

Rajbhandari S, Rasley J, Ruwase O, He Y. Zero: Memory optimizations toward training trillion parameter models. International Conference for High Performance Computing, Networking, Storage and Analysis, SC. 2020;2020-Novem: 1–24. doi:10.1109/SC41405.2020.00024

Snoek J, Larochelle H, Adams RP. Practical Bayesian Optimization of Machine Learning Algorithms. In: Pereira F, Burges CJ, Bottou L, Weinberger KQ, editors. Advances in Neural Information Processing Systems. Curran Associates, Inc.; 2012. Available: https://proceedings.neurips.cc/paper/2012/file/05311655a15b75fab86956663e1819cd-Paper.pdf

Biewald L. Experiment Tracking with Weights and Biases. 2020. Available: https://www.wandb.com/

Howard J, Gugger S. Fastai: A Layered API for Deep Learning. Information. 2020;11: 108. doi:10.3390/info11020108

Smith LN. Cyclical Learning Rates for Training Neural Networks. arXiv; 2017. Available: http://arxiv.org/abs/1506.01186

Smith LN, Topin N. Super-convergence: very fast training of neural networks using large learning rates. In: Pham T, editor. Artificial Intelligence and Machine Learning for Multi-Domain Operations Applications. SPIE; 2019. p. 36. doi:10.1117/12.2520589

Kingma DP, Ba J. Adam: A Method for Stochastic Optimization. arXiv; 2017. Available: http://arxiv.org/abs/1412.6980

Agarap AF. Deep Learning using Rectified Linear Units (ReLU). arXiv; 2019. Available: http://arxiv.org/abs/1803.08375

Misra D. Mish: A Self Regularized Non-Monotonic Activation Function. arXiv; 2020. Available: http://arxiv.org/abs/1908.08681

Loshchilov I, Hutter F. Decoupled Weight Decay Regularization. arXiv; 2019. Available: http://arxiv.org/abs/1711.05101

We have also corrected two references:

Before: 

Souza F, Nogueira R, Lotufo R. Portuguese Named Entity Recognition using BERT-CRF. arXiv preprint arXiv:190910649. 2019.

After:

Souza F, Nogueira R, Lotufo R. {BERT}imbau: pretrained {BERT} models for {B}razilian {P}ortuguese. 9th Brazilian Conference on Intelligent Systems, {BRACIS}, Rio Grande do Sul, Brazil, October 20-23 (to appear). 2020.

Before:

Lage-Freitas A, Allende-Cid H, Santana O, de Oliveira-Lage L. Predicting Brazilian court decisions. 2019; 1–4.

After:

Lage-Freitas A, Allende-Cid H, Santana O, Oliveira-Lage L. Predicting Brazilian Court Decisions. PeerJ Computer Science. 2022;8: e904. doi:10.7717/peerj-cs.904

B) REVIEWER 1

1. ABSTRACT - Sentence 3: This sentence does not motivate the need for the study as this study does not really help understand the steps of case flow.

We rephrased the sentence. See lines 27-28.

2. Line 8: machines  models

Done.

3. Line 7 - 9: We compare the predictive performance of the models to the predictions of twenty two highly skilled legal experts.

Done.

4. Line 10: ... of 0.3688 compared to 0.1253 from the human experts.

Done.

5. 2nd last sentence - Rewrite the conclusion to be more specific: Our results demonstrate that natural language machine learning techniques is a promising approach for predicting legal outcomes.

Done. Lines 33-35.

6. INTRODUCTION - Page 3: - Line 3 : admitted  submitted

Done.

7. First paragraph: "We propose the ..." Should specifically mention that the class of models used in this paper are deep learning, natural language processing (NLP) models.

Done. 

8. Last line: analysis conducted  predictions determined ..

Done. 

9. Second paragraph: "we evaluated ..."

We did not understand what the reviewer meant here.

10. Line 5: delete "we used"

Done.

11. Line 7: replace the first "architecture"  "model"

Done.

12. Third paragraph: "We discuss how ... "

We did not understand what the reviewer meant here.

13. Line 1: "an algorithm"  "models"

Done.

14. Sentence 1: Begin the paragraph with: AI-driven systems could have real potential to improve stability, predictability ... Adust the paragraph appropriately.

We rephrased the sentence. See lines 89-95.

15. While the benefits of such systems are described - it must be mentioned that much more work is required before automated systems replace human decision making.

Done. See lines 96-99.

16. INSTITUTIONAL BACKGROUD - This section must be reduced - while it may be usefull background - much of it is not necessary to understand the rest of the paper. It should rather be reduced from 5 pages to 2 or so pages.

We reduced this section from 2545 to 1700 words (a 33% decrease from the original size). We could not reach the size by you without losing too much information. We believe it is essential to put things in context by explaining where our models should be deployed. While there are many scientific papers about LSTMs and Transformers in English, there is not enough material about the Brazilian legal system. As legal researchers, we feel obliged to provide this information to our readers, as we do not expect our audience to understand how Brazilian courts work.

17. NLP MODELS - The paper should include a background on the NLP models used in the study. For e.g. LSTM and Transformer models should be described. 

Done. See lines 234-342.

18. METHODOLOGY - Paragraph Two: clarify if any pre-processing was done - either no preprocessing, or if any then what did this entail.

Done. See lines 363-366.

19. Paragraph Three: did each participant predict each of the 690 appeals?

Done. See lines 369-390 and 609.

20. In what language are the full text written in?

Done. See lines 366-367.

21. Page 13 - first two paragraphs - would better fit in the introduction that discusses the motivation for the study.

Done. See lines 64-77.

22. Page 14: While the labelling process is sensible, it is not fool proof. Where there any processes used to verify the labels? For e.g. sampling some of the decisions?

Done. See lines 448-449.

23. Page 16: First paragraph in "Experimental setup" ".. to all parties in production. "  "... to all parties."

Done.

24. Experimental Setup: The training set up of the different models should give the hyper parameters used and explain how these were tuned.

Done. See lines 510-514 and 514-589.

25. Some more details of the model set up would be useful if they differed from the standard set up discussed in the literature.

Done. See lines 510-589, especially (new) tables 2, 3, and 4.

26. Legal expert's analysis"... 17 highly skilled ..." This number contradicts the numbers given earlier in the paper - where 22 is reported. Please fix.

Done.

27. Page 19: While we understand the difficulty to obtain human expert decisions,it would have good to evaluate inter-human agreement i.e. if participants were to have evaluated the same cases.

We understand and agree with your point. However, we knew how busy our subjects were with their everyday court activities. Therefore, asking them to label the same data point more than once would be risky, resulting in less labeled data on our human expert’s dataset, weakening the comparison between humans and ML models (our primary goal).

28. RESULTS - The hyperparameters must be clearly specified in the methods. 

Done. Please see tables 2, 3 and 4 (lines 531, 563, and 588)

29. DISCUSSION - Paragraph Two: Line 2: "example The"  "examople. The"

Done.

33. Paragraph Four: Sentence Two is not substantiated: How will the tool help to detect inconsistencies - as the tool does not, as described, cluster cases on similarity. So this entire paragraph is problematic and should be left out or substantiated carefully. 

We removed this part.

Lastly, we want to thank you for the thorough review of our work.

C) REVIEWER 2

1.The paper provides an interesting contribution to the literature. I believe that the paper could improve the explanation of the models that are used and provide the codes so other researchers can follow these steps. It could open an important reseach area applied to judicial sentences and increase the potential for citations of the paper. 

Thanks for your input. We provided additional information about the architectures and hyperparameters used. See lines 510-589, especially (new) tables 2, 3, and 4. All code is available at https://github.com/eliasjacob/paper_brcad5

---

## [Editor Report · Decision Letter 1]

18 Jul 2022

Using deep learning to predict outcomes of legal appeals better than human experts: A study with data from Brazilian federal courts

PONE-D-22-07895R1

Dear Dr. Jacob de Menezes-Neto,

We’re pleased to inform you that your manuscript has been judged scientifically suitable for publication and will be formally accepted for publication once it meets all outstanding technical requirements.

Kind regards,

Donrich Thaldar

Academic Editor

PLOS ONE
---

## [Editor Report · Acceptance letter]

20 Jul 2022

PONE-D-22-07895R1 

Using deep learning to predict outcomes of legal appeals better than human experts: A study with data from Brazilian federal courts 

Dear Dr. Jacob de Menezes-Neto:

I'm pleased to inform you that your manuscript has been deemed suitable for publication in PLOS ONE. Congratulations! Your manuscript is now with our production department. 

Kind regards, 

on behalf of

Professor Donrich Thaldar 

Academic Editor

PLOS ONE